# In-Situ and Ex-Situ Characterization of Femtosecond Laser-Induced Ablation on As_2_S_3_ Chalcogenide Glasses and Advanced Grating Structures Fabrication

**DOI:** 10.3390/ma12010072

**Published:** 2018-12-26

**Authors:** Hongyang Wang, Dongfeng Qi, Xiaohan Yu, Yawen Zhang, Zifeng Zhang, Tiefeng Xu, Xiaowei Zhang, Shixun Dai, Xiang Shen, Baoan Song, Peiqing Zhang, Yinsheng Xu

**Affiliations:** 1Laboratory of Infrared Materials and Devices, The Research Institute of Advanced Technologies, Ningbo University, Ningbo 315211, China; why_run@163.com (H.W.); 17794615469@163.com (X.Y.); zywsara1026@126.com (Y.Z.); xutiefeng@nbu.edu.cn (T.X.); zhangxiaowei@nbu.edu.cn (X.Z.); daishixun@nbu.edu.cn (S.D.); songbaoan@nbu.edu.cn (B.S.); zhangpeiqing@nbu.edu.cn (P.Z.); xuyinsheng@nbu.edu.cn (Y.X.); 2College of Mechanical and Electronic Engineering, Chaohu University, Hefei 230000, China; zhangzfxmu@hotmail.com

**Keywords:** laser processing, femtosecond laser

## Abstract

Femtosecond laser pulse of 800 nm wavelength and 150 fs temporal width ablation of As_2_S_3_ chalcogenide glasses is investigated by pump-probing technology. At lower laser fluence (8.26 mJ/cm^2^), the surface temperature dropping to the melting point is fast (about 43 ps), which results in a clean hole on the surface. As the laser fluence increases, it takes a longer time for lattice temperature to cool to the melting point at high fluence (about 200 ps for 18.58 mJ/cm^2^, about 400 ps for 30.98 mJ/cm^2^). The longer time of the surface heating temperature induces the melting pool in the center, and accelerates material diffusing and gathering surrounding the crater, resulting in the peripheral rim structure and droplet-like structure around the rim. In addition, the fabricated long periodic As_2_S_3_ glasses diffraction gratings can preserve with high diffraction efficiency by laser direct writing technology.

## 1. Introduction

Chalcogenide glasses (ChGs) have gained extensive interest due to its wide transparency range, high refractive index, high nonlinear optical coefficient and high photosensitivity [1,2,3,4]. In this case, ChGs can be used as excellent candidate materials for optical communication, optical sensing and optical recording areas [5,6]. In particular, the generation in surface nano-structures and nanohole arrays or nano-gratings with the largest achievable manipulation of refractive index, master the key to functional devices in future optical systems [7,8,9]. Among these manufacturing technologies, femtosecond lasers has opened up new applications and possibilities in the bulk glass materials due to its limited heat affected zone, very high flexibility and noncontact process, which has been used in the manufacturing of Nd:LuVO_4_, Ge-Sb-Se,As_2_Se_3_, Tm:YVO_4_ and Nd:GLSG (neodymium doped gallium lanthanum sulfide glass) [10,11,12,13,14]. Femtosecond direct writing methods in the case of chalcogenide bulk glass and fibers have been used to make 3D holographic recordings, bulks of gratings and waveguides [15,16,17,18]. In addition, long-period gratings are an important part of optical communication systems for optical filtering and mode conversion, optical gain, and sensing applications. [19,20]. For example, the long periodic gratings have nonlinear periodic structure embodiment; in this condition, the nonlinear refractive index is used as optical switching devices [21].

Basic understanding of the process of femtosecond laser-induced material changes is important for the prediction and optimization of laser processes [22,23]. At present, optical detection technology has been used in laser–material interaction research [24,25]. For example, Pump-probing and time-resolved shadow-graphics techniques have been used to directly observe ablation processes in very short (picosecond) timescale [26,27,28,29]. At present, there were few studies on the surface morphology evolution of femtosecond laser-induced chalcogenide glass materials, and the mechanism of the surface morphology evolution is still not perfectly explained. Nevertheless, even in the simplest case of chalcogenide materials, such as As_2_S_3_ and As_2_Se_3_, the femtosecond laser-induced process of phase change and the ablation have not been investigated. What is more important is that basic understanding of the process of femtosecond laser-induced processes is important for the prediction of laser manufacturing.

In this letter, we study the evolution process of surface morphology of femtosecond laser-induced As_2_S_3_ chalcogenide glasses, and the basic structural property of the As_2_S_3_ has been reported in some references [30]. We have carried out pump-probing technology to investigate the femtosecond laser-induced ablation processes in As_2_S_3_ glass. The pump-probing setup elucidates the transient breakup of the edge rim, droplet-like and ablation area. By studying the relationship between the surface morphologies and the laser fluences, we can fabricate a long-period grating structure with smooth morphology on the surface of As_2_S_3_.

## 2. Materials and Methods

The pump-probing setup revealed the evolution of different morphologies during the ablation processes. The pump-probing imaging system was set up as shown in Figure 1. Ti Sapphire laser pulses of 800 nm wavelength and 150 fs temporal width impinged As_2_S_3_ targets. The laser beam was focused by a × 5 corrected, non-achromatic long working distance objective lens at normal incidence. For the reflection probing system, the 632.8 nm He-Ne continuous laser has been focused onto the center of the irradiated region at normal incidence. The As_2_S_3_ specimens are positioned at the focal plane of the probing laser whose location is defined by knife-edge beam profiling. The laser processing beam and probe beam are also measured under the knife-edge method. The intensity of the reflection probe signal was measured by a fast photodiode coupled to an oscilloscope. The oscilloscope is used to record the actual delay time of the processing laser signal and the probing laser signal. To ensure true representation, at least four signals are examined at each delay setting.

## 3. Results and Discussion

Typical submicron-scale structures are shown in SEM and AFM images. Smooth crater structures can be formed after irradiation of laser at lower laser fluence (8.26 mJ/cm^2^) on As_2_S_3_ glass surface, and rim structure around the crater appears as the increasing of the laser fluences, as shown in Figure 1(a1–a3). Figure 1(b1–b3) give detailed information of edges of the surface crater structures, and the outer edge is relatively smooth at low laser fluence (8.26 mJ/cm^2^). As the fluence of laser increases to 18.58mJ/cm^2^, rim structure combined with some droplet-like structures on the edge appear, and the size of these droplet-like structures grows bigger around the crater outskirt as the increasing of the laser fluence (30.98 mJ/cm^2^). The detailed crossing-sectional characters of the crater structures are measured by the AFM images. At higher laser fluence, as shown in Figure 1(d1–d3), the rim structure round the crater appears and the height of the rims becomes larger.

Next, the detailed relationship between the characters of the surface craters and the laser fluences are investigated, which is shown in Figure 2. The red data points show the measured crater depth as a function of fluence, and the red curve is the corresponding simulated result. For the femtosecond laser irradiation, the OPA (one-photon absorption) should be taken into account to explain the ablation depth. Besides, laser fluences described by a Lambert–Beer, the balance equation for carrier number density N, the complex refractive index n of the material should be considered [31]:

Lambert–Beer law:(1)$$\frac{\partial I}{\partial z}=-(\alpha_{0}+\alpha_{Drude})I-\beta I^{2}$$

Carrier number density N of the balance equation:(2)$$\frac{\partial N}{\partial t}+\nabla \cdot (-D_{0}\nabla N)=\frac{\alpha_{0}I}{h\omega }+\frac{\beta I^{2}}{2h\omega }$$

The complex refractive index n is calculated by the Drude model:(3)$$n=\sqrt{\varepsilon_{As_{2}S_{3}}-\frac{\omega_{p}^{2}}{\omega^{2}+i\omega /\tau_{d}}}$$

In Equation (1), *I* = (1 − *R*)*I*_0_, *R* is the surface reflectivity. For the As_2_S_3_ materials, the OPA coefficient *α*_0_~10^3^/cm [32,33,34], $\varepsilon_{As_{2}S_{3}}$ is the dielectric constant of As_2_S_3_ [35], *ω* is the angular frequency of the pulse, and $\tau_{d}$ is the damping time (1.1 fs). The plasma frequency *ω_p_* = (4*πNe*^2^/*m**)^1/2^, and electron effective mass *m** = 0.18 m_e_. The absorption of the incident laser in the plasma *α_Drude_* = 4πk/λ, *D*_0_ is the coefficient of ambipolar diffusivity (18 cm^2^/s) and hω is the photon energy. A single shot ablation threshold *F_th_* = 7.21 mJ/cm^2^ can be measured. In addition, we also determine *F_th_* by measuring the crater diameter *D* for different laser fluences and by using the linear relationship *D*^2^ = 2*r_f_*^2^[*ln*(*F*) − *ln*(*F_th_*)] [36], where the 1/*e* beam radius, r_f_, is about 18 μm. The black data points show the hole diameter as a function of laser intensity, and the As_2_S_3_ ablation threshold (*F_th_*) is estimated as 7.19 mJ/cm^2^, which is consistent with the former result (*F_th_* = 7.21 mJ/cm^2^). Finally, the aspect ratio (color in blue), the crater diameter divide by the depth, is also described. As the laser pulse energy increases, the aspect ratio first increases and then decreases. The largest aspect ratio is 156 at 12.39 mJ/cm^2^ of laser fluence, which is related to the competition mechanism between the melting depth and Gaussian laser beam distribution, and the melting depth grows directly with the laser fluence and reaches the saturation state [37].

The reflection probing reveals the transient dynamics of laser interaction with As_2_S_3_, which is shown in Figure 3. The drop in low fluence can be concluded as the effect of the ablation since the ablation gives a crater in the irradiation area, indicating a weaker reflection. And under high fluence pump laser irradiation, the reflections directly drop to a lower state. Therefore, the reflection measurement validates the dynamics of ablation processing. Besides, the trends of the surface temperature after laser irradiation are also investigated, as shown in Figure 3. The energy conversion follows a one-dimensional dual temperature model which was proposed by Qiu and Tien [38].
(4)$$C_{e}\frac{\partial }{\partial t}T_{e}=\frac{\partial }{\partial t}k\frac{\partial }{\partial x}T_{e}-G(T_{e}-T_{l})+S$$
(5)$$C_{l}\frac{\partial }{\partial t}T_{l}=G(T_{e}-T_{l})$$
(6)$$S=0.94\frac{1-R}{t_{p}\delta }J\cdot \exp\left[-\frac{x}{\delta }-2.77{\left(\frac{t}{t_{p}}\right)}^{2}\right]$$

In this equation, *C_e_*, is the electron heat capacity and *C_l_* is the lattice heat capacity. *G* is the electron-lattice coupling factor, *S* is the radiation heating source term, and R is the reflectivity, *δ* is the radiation penetration depth, *J* is the energy of laser pulse [39]. All the material simulated is As_2_S_3_ glass and its physical constants are listed in Table 1, and the initial and boundary conditions for both the electron and the lattice systems can be defied as $T_{e}\left(x,-2t_{p}\right)=\;T_{l}\left(x,-2t_{p}\right)=T_{0}$. At low laser fluences, the energy of these heated electrons can be transferred to the surrounded lattice very rapidly (several picoseconds), as shown in Figure 4.

We can find that the surface temperature reaches a maximum value of 697 K at 6 ps, and the melting depth is 68 nm when the laser fluence is 8.26 mJ/cm^2^. With the increase of laser fluence, the surface temperature can reach 969 K and 1246 K after 7 and 8 ps laser irradiation, as shown in Figure 4. Besides, the melting depths are 161 nm and 252 nm, respectively. The depths obtained by experiments at the same three fluences are 72 nm, 162 nm and 262 nm, which are consistent with the theory results. For the different laser fluences, it takes about 43 ps, 173 ps and 450 ps for the temperature dropping to the melting point of the material, which is consistent with reflection probing experimental results, as shown in Figure 3. 

At lower laser fluence, as shown in Figure 3, and the non-thermal mechanisms play a major role for the laser–materials processing, which results in a clean hole on the surface. As the laser fluence increases, firstly, lattice temperature reaches a maximum (7 ps) and then the lattice temperature begins to gradually decline. Since it takes longer for the lattice temperature to drop to the material melting point at high fluence, with increasing laser fluence, it takes much longer for the surface temperature to drop to the melting temperature, which is due to the theory of the one-dimensional dual temperature model. In this condition, the laser is the only heating source and the higher laser fluence induces the higher surface temperature; it takes more time to diffuse the surface heat. The material surface reflectivity decreases more slowly (about 200 ps for 18.58 mJ/cm^2^ in Figure 3). The time of the surface heating temperature can reach 200 ps or above, causing the melting pool in the center, resulting in the material diffuses and gathers surround the crater. For higher laser fluence (above 30.98 mJ/cm^2^), the liquid As_2_S_3_ materials are rapidly pulled out of the pool, resulting the peripheral rim structures.

The former results show that holes can be fabricated at the fluence of 8.26 mJ/cm^2^. In addition, direct laser writing (DLW) in processing is a fast and flexible method for long periodic grating fabrication [29]. Figure 5 shows the laser direct writing single gratings (a–c) and the composite gratings (d–f) at the laser scanning velocity of 1, 2 and 5 mm/s, respectively. In the optical microscope pattern, black color areas are the laser direct writing areas, and gray areas are single grating (a1–c1) and the composite grating structures (d1–f1) with a period of 48, 35 and 20μm, respectively. The depths of these gratings are about 3.8, 2.6, 1.1 μm, respectively. For the lower scanning velocity, the morphology of the gratings is not perfect, which result from the multiple-pulse laser acting on the surface, and thereby forming some rim structure around the grating, as shown in Figure 5(a2,d2). For the larger scanning velocity, it is equivalent to a single-pulse laser irradiation effect, and wrinkled structures are formed around the grating, as shown in Figure 5(c1,f1).

In order to investigate the quality of the grating, we measured the diffraction efficiency of single gratings, as shown in Figure 5(a3–c3) and composite gratings (d3–f3). The laser fluence of the light source with a wavelength of 632.8 nm is 2.3 mw and the distance between the grating and the screen is 60 cm, and the incident angle of the laser through the grating is θ = 0°. The diffraction efficiency of long periodic gratings is η + 1 = 3.48% for d = 20 μm, η + 1 = 6.30% for d = 35 μm, and η + 1 = 5.74% for d = 48 μm. And, efficiency values of (η + 2) are 0.91%, 1.71%, and 1.52%, respectively. The diffraction efficiency of composite gratings is η + 1 = 1.74% for d = 20 μm, η + 1 = 3.17% for d = 35 μm, and η + 1 = 2.70% for d = 48 μm. The diffraction efficiency of the grating increases first and then decreases with the increases of laser scanning velocity, which results from the morphologies of the grating structures. The diffraction efficiency of the grating increases and then decreases with the increases of laser scanning velocity, which results from the morphologies of the grating structures. The grating structures with such wrinkled or rim structures around gratings and the photo-darkening process [43,44] can scatter or absorb the incident light, finally reducing the diffraction efficiency of the grating structures. In this condition, the long periodic gratings on As_2_S_3_ glasses surface can preserve high diffraction efficiency with a wide range (0.6–8 μm).

## 4. Conclusions

In conclusion, we have carried out pump-probing technology to investigate the femtosecond laser-induced ablation processes in As_2_S_3_ glass. The pump-probing setup elucidates the transient breakup of the edge rim, droplet-like and ablation area. Besides, diffraction gratings with period of 20, 35, and 48 μm on the As_2_S_3_ chalcogenide glass surface are fabricated with different laser scanning velocity. And the first-order diffraction efficiency of the gratings to be up to 6.3% at λ (632.8 nm) with transmittance operation at normal incidence can be fabricated at a proper laser scanning velocity and laser fluence.

## Figures and Tables

**Figure 1 materials-12-00072-f001:**
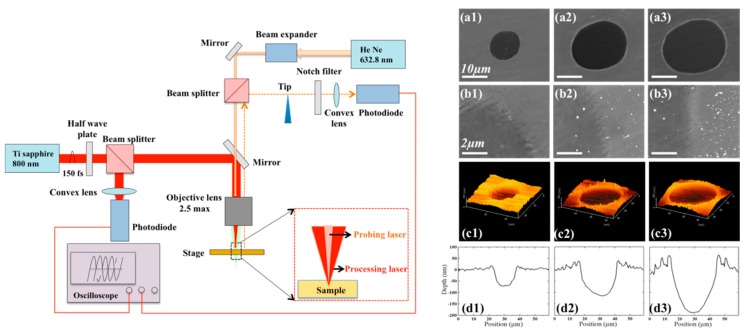
The left image, pump-probing setup of femtosecond laser-induced in As_2_S_3_, and the fullwidthathalfmaximum (FWHM) of pump laser and probing laser are 36 μm and 20 μm, respectively. (**a1**)–(**a3**) The right image, scanning electron microscopy; (**b1**)–(**b3**) the magnified SEM images; (**c1**)–(**c3**) AFM images; (**d1**)–(**d3**) and the cross-sections AFM images. Irradiation laser flences: 8.26 mJ/cm^2^, 18.58 mJ/cm^2^ and 30.98 mJ/cm^2^, respectively.

**Figure 2 materials-12-00072-f002:**
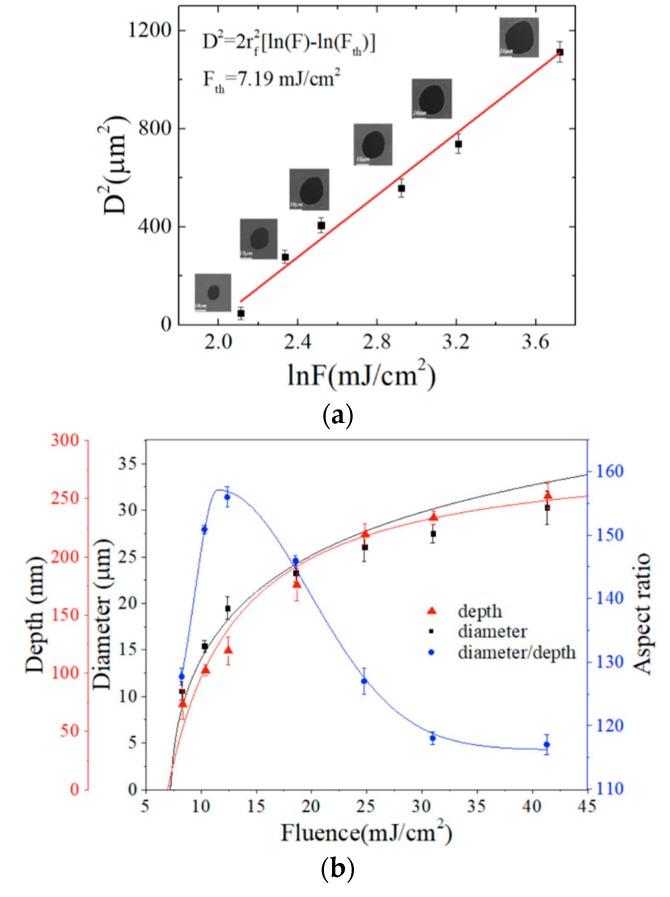
(**a**) The ablation diameter of *r_a_*^2^, versus the laser fluence ln(*F*). (**b**) The data points in red, black and blue indicate the hole depth, the diameter and the corresponding aspect ratio, respectively, the red solid curve, the black solid curve and the blue solid curve show the calculated hole depth, the diameter and the aspect ratio using a 150-fs pulse duration laser, respectively.

**Figure 3 materials-12-00072-f003:**
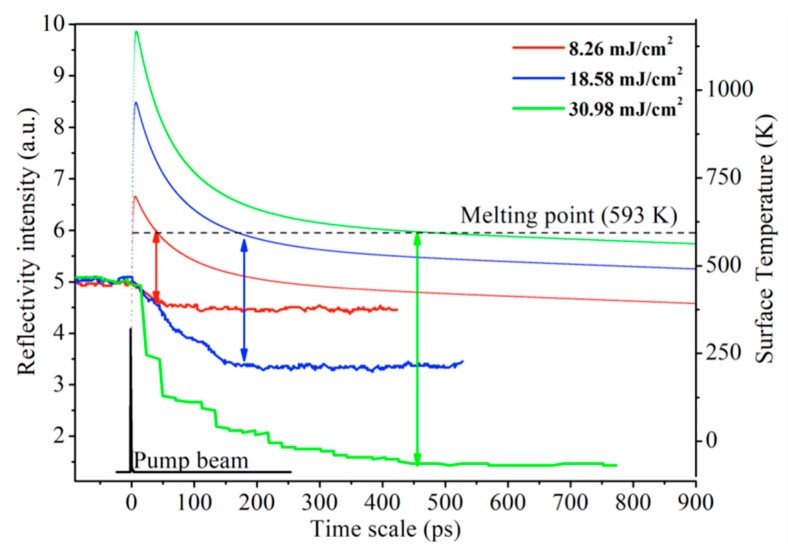
The pump-probing reflectivity signals of laser-induced ablation of As_2_S_3_ with different laser fluences in scatter-lines, and the corresponding simulated surface temperature as a relationship of time for the applied incident laser intensity in solid lines.

**Figure 4 materials-12-00072-f004:**
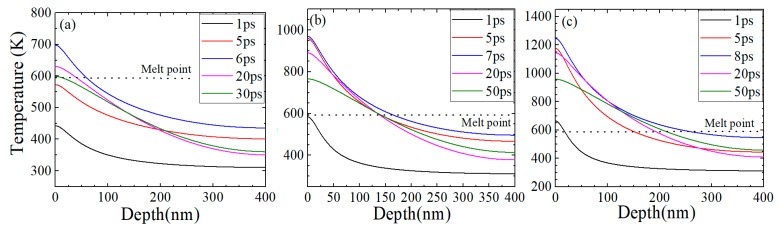
The relationship between surface temperature and depth of the material at different delay times, (**a**) 8.26 mJ/cm^2^, (**b**) 18.58 mJ/cm^2^, (**c**) 30.98 mJ/cm^2^, respectively.

**Figure 5 materials-12-00072-f005:**
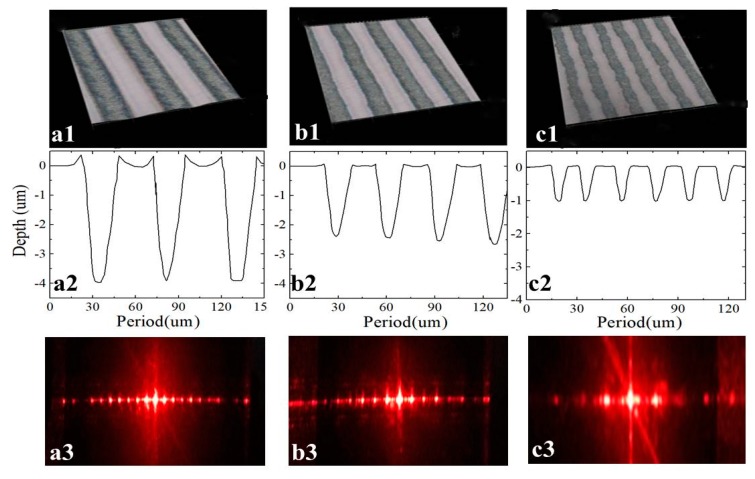
The laser energy was 8.26 mJ/cm^2^, the scanning speed were 1, 2 and 5 mm/s, and the corresponding period were 48, 35 and 20μm single gratings (**a**–**c**) and composite gratings (**d**–**f**). (**a1**–**f1**) were the pattern of optical microscope, (**a2**–**f2**) were the step meter profile, and (**a3**–**f3**) were the grating diffraction pattern.

**Table 1 materials-12-00072-t001:** Parameters forAs_2_S_3_ used in heat calculation [38,39,40,41,42].

As_2_S_3_ (Parameters)	Values
Initial temperature (*T*_0_)	300 K
Thermal conductivity (*k*)	0.17 W·m^−1^·C^−1^
Lattice heat capacity (*C_i_*)	1 × 10^6^ J·m^−3^·K^−1^
Electron heat capacity (*C_e_*)	502 J·Kg^−1^·K^−1^
Electron-phonon coupling factor (*G*)	2.6 × 10^16^ W·m^−3^·K^−1^
Reflection coefficient (*R*)	0.6
Radiation penetration depth (*δ*)	15.3 nm

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
