# Peer review of "In-Situ and Ex-Situ Characterization of Femtosecond Laser-Induced Ablation on As2S3 Chalcogenide Glasses and Advanced Grating Structures Fabrication"

_materials, 2018, doi:10.3390/ma12010072_

Reviewer 1 Report

COMMENTS TO AUTHORS

Your paper entitled “In-situ and ex-situ characterization of femtosecond laser-induced ablation on As2S3 chalcogenide glasses and advanced grating structures fabrication” deals with the influence of the laser fluence on the surface of the glass. In general point of view, your manuscript exhibits a well-defined skeleton. However, I have some comments:

1) Your work is devoted to the specific glass As2S3. Perhaps a paragraph in the introduction mentioning some papers in which the fs laser photowritting is carried out to modify the refractive index of the glass would be appreciate. As a matter of fact you mention different works but only a few are related to As2S3.

2) In the introduction you could also write why the fabrication oflong-period grating structure for infrared range is so important by given some examples.

3) From line 192 to 206 you discuss the quality of the created gratings. Could you position your results compared to analogous experiments in other type of glasses (I mean for either chalcogenide or oxide glasses)

4) The manuscript from line 101 to 112 is a little bit complicated to well understand, mainly because the paragraph is badly written. For example : “The plasma frequency ωp  = (4πNe2/m*)1/2,  where m* =0.18me ”. It’s not a sentence. Also, other equations are directly inserted in the text without clear explaination. Authors should improve the quality of this part to allow a better understanding for the reader.

5) Lines 138/139, you write “chalcogenide glass” but you clearly mention the composition As2S3.

6) Some typographic errors are in the text yet. Please read carefully your manuscript and correct them : for instance lines 45, 56, 82, 113, 114, …

Finally, I will recommend minor revisions  for your submitted paper.

Author Response

Manuscript ID: materials-396423

TITLE: In-situ and ex-situ characterization of femtosecond laser-induced ablation on As2S3chalcogenide glasses and advanced grating structures fabrication

We are grateful to the reviewer for the comments on our manuscript (Manuscript ID: materials-396423). Detailed point-by point response to the comments in the attachment file:

Reviewer 2 Report

In this article the authors showed in-situ and ex-situ characterization of femtosecond laser induced ablation. 

The theory behind the laser irradiation is not clear and not properly described.

Why did they chose to use 800 nm for investigation. 

What is the reason behind considering one-photon absorption and two-photon absorption using 800 nm wavelength. The authors just used them in the equation.

If one-photon absorption occurs then we can't have two-photon absorption.

The authors should explain above points for acceptance of the article

Author Response

(The authors gave the same response as above.)

Reviewer 3 Report

Femtosecond laser structuring of chalcogenide glass was monitored with pump probe method. Numerical simulations of light matter interaction are also presented. The two-photon absorption cross section was directly  measured with Z-scan for the conditions similar to the  described experiment and the value was ten times larger as used in modeling (Optics Express Vol. 14, Issue 17, pp. 7751-7756 (2006)). The melting and ablation threshold have to be measured directly. The fig 2 has to  be presented in diameter^2 vs ln(fluence)  scale to obtain linear slopes. this can reveal difference in energy deposition. Capillary driven molten flows were earlier reported (Juodkazis et al 2006 Nanotechnology 17 4802). Why low diffraction efficiency was observed. Was it related to the photo-darkening (Optics Express Vol. 14, Issue 17, pp. 7751-7756 (2006))? typos: ...of As2S3 glasses can preserve with high diffraction efficiency by laser direct writing technology. As2S3andAs2Se3, fullwidthathalfmaximum, As2S3atsingle-shot, 12.39mJ/cm2of, energy, By extrapolating

Author Response

Manuscript ID: materials-396423

TITLE: In-situ and ex-situ characterization of femtosecond laser-induced ablation on As2S3chalcogenide glasses and advanced grating structures fabrication

We are grateful to the reviewer for the comments on our manuscript (Manuscript ID: materials-396423). Detailed point-by point response to the comments in the attachment file:

Round  2

Reviewer 2 Report

Thank you to the authors for answering my questions. 

I would suggest please add, why does it take longer time for the lattice temperature to drop in high laser fluence case. It no where clear inside the manuscript.  Also please emphasize you are considering a crystalline material and give a reference which has provided the structural property of the AS2S3.

Author Response

Dear reviewer,

we have response point by point and list it in the attachment file.
